

# Competing Transfer Pathways in Direct and Indirect Dynamic Nuclear Polarization MAS NMR Experiments on HIV-1 Capsid Assemblies: Implications for Sensitivity and Resolution

Ivan V. Sergeyev[1], Caitlin M. Quinn[2], Jochem Struppe[1], Angela Gronenborn[3,4*], Tatyana Polenova[2,3*]

[1]Bruker Biospin Corporation, 15 Fortune Drive, Billerica, Massachusetts 01821, United States
[2]Department of Chemistry and Biochemistry, University of Delaware, Newark, Delaware 19716, United States
[3]Pittsburgh Center for HIV Protein Interactions, University of Pittsburgh School of Medicine, 1051 Biomedical Science Tower 3, 3501 Fifth Avenue, Pittsburgh, Pennsylvania 15261, United States
[4]Department of Structural Biology, University of Pittsburgh School of Medicine, 3501 Fifth Ave., Pittsburgh, PA 15261, United States

*Correspondence to*: Tatyana Polenova (tpolenov@udel.edu), Angela M. Gronenborn (amg100@pitt.edu)

**Abstract.** Dynamic nuclear polarization-enhanced (DNP) magic angle spinning (MAS) NMR of biological systems is a rapidly growing field. Large signal enhancements make the technique particularly attractive for signal-limited cases, such as studies of complex biological assemblies or at natural isotopic abundance. However, spectral resolution is considerably reduced compared to ambient-temperature non-DNP spectra. Herein, we report a systematic investigation into sensitivity and resolution of 1D and 2D $^{13}$C-detected DNP MAS NMR experiments on HIV-1 CA tubular assemblies. We show that the magnitude and sign of signal enhancement as well as the homogeneous line width are strongly dependent on the biradical concentration, the dominant polarization transfer pathway, and the enhancement buildup time. Our findings provide guidance for optimal choice of sample preparation and experimental conditions in DNP experiments.

## 1 Introduction

Since the seminal reports of DNP-enhanced MAS NMR experiments in biological systems by Griffin and coworkers in the early 2000s (Bajaj et al., 2003;Rosay et al., 2003;Rosay et al., 2001), the field has evolved rapidly. Commercial instrumentation operating at magnetic fields from 9.4 T to 21.1 T has enabled the use of DNP technology by multiple NMR groups across the globe. Today, DNP MAS NMR is used in structural studies of a broad range of biological systems, including soluble proteins (Jeon et al., 2019), amyloid fibrils and nanocrystals (Bayro et al., 2011;Debelouchina et al., 2013;Frederick et al., 2017;van der Wel et al., 2006;Debelouchina et al., 2010), membrane proteins (Rosay et al., 2003;Cheng and Han, 2013;Tran et al., 2020;Salnikov et al., 2017;Smith et al., 2015) (Wylie et al., 2015), nucleic acids (Wenk et al., 2015), viruses and viral protein assemblies (Gupta et al., 2016;Jaudzems et al., 2018;Gupta et al., 2019;Lu et al., 2019;Rosay et al., 2001;Sergeyev et al., 2011), biomaterials (Koers et al., 2013;Ravera et al., 2015;Viger-Gravel et al., 2018), unfolded and misfolded proteins (König et al., 2019), and intact cells (Viennet et al., 2016;Albert et al., 2018;Judge et al., 2020;Yamamoto



et al., 2015), including at natural isotopic abundance (Viger-Gravel et al., 2018;Takahashi et al., 2012). The key advantage of DNP-enhanced MAS NMR is the tremendous sensitivity enhancements afforded by the transfer of polarization from electron spins to nuclear spins. Sources of electron spins are either endogenous paramagnetic groups in the molecule (Maly et al., 2012) or externally added paramagnetic species (Hu et al., 2004). Theoretical maximum DNP enhancement factors, ε, are ~660 ([1]H),

2,624 ([13]C), and 6,511 ([15]N), with ε=250 measured as the highest [1]H enhancement for a biological molecule (Wenk et al., 2015). Not unexpected, experimental DNP enhancements in biological systems vary widely, depending on the magnetic field strength, the type and concentration of radical/biradical, the temperature, the MAS frequency, and the nature of the sample. The largest gains have been observed at moderate magnetic fields (9.4-14.1 T), such as ε=148 for direct [13]C excitation in perdeuterated microcrystalline SH3 (Akbey et al., 2010) and ε=64-100 for CP-based [13]C and direct [19]F excitation in fully

protonated HIV-1 capsid tubes (Gupta et al., 2016;Gupta et al., 2019;Lu et al., 2019). On the contrary, modest gains have been detected for membrane proteins with ε=4-10 (Wylie et al., 2015) and at higher magnetic fields of 18.8 T (Gupta et al., 2016). Recently, large DNP signal enhancements ε=35-41 have been reported on a histidine impregnated with the biradical HyTEK2 in 1,1,2,2-tetrachloroethane at a magnetic field of 21.15 T (Berruyer et al., 2020), although to date soluble DNP polarization agents suitable for experiments in biological systems have not yet been reported.

The sensitivity gains offered by DNP open new avenues for characterization of biological systems intractable by conventional MAS NMR techniques. Unfortunately, these gains are often accompanied by a loss of resolution (Can et al., 2015;Geiger et al., 2016). At cryogenic temperatures of 120 K and lower, where signal enhancements are the highest, severe broadening of spectral lines is common, limiting widespread applications to biological systems. Sources for reduced resolution are varied, caused by either freezing-out different conformational substates (Linden et al., 2011), paramagnetic broadening

(Rogawski et al., 2017;Gupta et al., 2016), or both. Interestingly, some systems, such as HIV-1 capsid assemblies (Gupta et al., 2016;Gupta et al., 2019), bacterial T3SS needles formed by MxiH protein (Fricke et al., 2016), Pf1 phage (Sergeyev et al., 2017), and Acinetobacter phage 205 capsid (Jaudzems et al., 2018) yield well-resolved DNP MAS NMR spectra; for others, such as disordered systems, the resolution is very poor (reviewed in (König et al., 2019)).

   While it appears that large signal enhancements and high spectral resolution can be observed in DNP experiments of

ordered rigid systems, a systematic understanding of sample and experimental conditions required to attain maximum sensitivity and resolution is lacking. One important consideration in the sample preparation relates to the concentration of the paramagnetic polarization agent, most commonly a biradical such as TOTAPOL (Song et al., 2006) or AMUPol (Sauvée et al., 2013). For most biological studies to date biradical concentrations range from 8 to 28 mM (reviewed in (Jaudzems et al., 2019)). Also of critical importance is the DNP polarization transfer pathway (Thankamony et al., 2017;Aladin and Corzilius,

2019, 2020). Although distinct polarization transfer pathways have been discussed to underlie the different DNP mechanisms, we will restrict discussion here to only the three-spin-flip mechanism known as the cross-effect (CE) (Hu et al., 2004) due to its unique position in the DNP field profile. Notably, even CE DNP encompasses several possible transfer pathways, each with a unique signature. In the indirect pathway, the electron polarization is first transferred from the radicals' electrons to the surrounding protons, followed by spin diffusion throughout the proton spin network and subsequent transfer of the proton



polarization to the nucleus of interest by cross polarization (CP). This pathway is exploited in most of the DNP experiments to date and yields high signal enhancements with short buildup times and relatively narrow lines, unaffected by paramagnetic broadening (Aladin and Corzilius, 2020). The direct DNP process entails coherent transfer of electron polarization to the nucleus of interest without any involvement of the proton spin network. Its drawbacks are significant paramagnetic broadening and low efficiency due to its slow spread (Aladin and Corzilius, 2020). However, as an advantage, direct DNP experiments

permit the detection of sites close to the paramagnetic centers. Simultaneously with the direct DNP effect, an indirect, incoherent DNP transfer, dubbed SCREAM-DNP, may occur, driven by molecular motion-associated heteronuclear cross-relaxation (Aladin and Corzilius, 2019). This pathway results in small negative signal enhancements, typically on or around mobile groups such as methyls.

      Herein, we report on a systematic study of $^{13}$C DNP signal enhancements and spectral resolution for tubular assemblies

of HIV-1 CA capsid protein. These CA capsid assemblies have been extensively characterized in our laboratory, including by $^{13}$C and $^{19}$F DNP-enhanced MAS NMR (Lu et al., 2019;Gupta et al., 2019;Gupta et al., 2016). For the experiments described here, the samples were prepared with 4.3, 22.8, and 28.2 mM AMUPol. All three polarization transfer pathways were detected: indirect, direct, and SCREAM-DNP. Overall, the dominant pathway is determined by the biradical concentration and DNP signal buildup time. The magnitude and sign of the signal enhancements as well as line widths are strongly dependent on the

biradical concentration and the polarization transfer pathway. 89- and 6.4- fold $^{13}$C signal enhancements were detected in CP and direct polarization (DP) experiments. Our findings also reconcile the large variations in signal enhancements and resolution reported by different research groups.

      Taken together, the results presented and discussed here are exciting and indicate that it is possible to select a desired DNP transfer pathway, and hence the information content of the spectra, by a judicious choice of sample preparation and

experimental conditions, thus further highlighting the unique capabilities of DNP-enhanced MAS NMR applications for structural biology.

## 2 Materials and methods

### 2.1 Samples

      5F-Trp,U-$^{13}$C,$^{15}$N-labeled CA (NL4-3 strain) was expressed and purified as described in our previous report (Lu et al.,

2019). Tubular assemblies of CA were prepared from 30 mg/mL protein solutions in 25 mM phosphate buffer (pH 6.5) containing 2.4 M NaCl, by incubation at 37 ºC overnight. The DNP samples were prepared following our previously established protocols (Gupta et al., 2016). In brief, the biradical, AMUPol (15- {[(7-oxyl-3, 11-dioxa-7-azadispiro [5.1.5.3] hexadec- 15-yl) carbamoyl] [2-(2,5,8,11-tetraoxatridecan-13-ylamino)} - [3,11-dioxa-7-azadispiro [5.1.5.3] hexadec-7-yl]) oxidanyl) (Sauvée et al., 2013) was added to 11.0, 11.7, and 11.6 mg of pelleted tubes. To dissolve the AMUPol, the pellets

were gently stirred. 20% (v/v) glycerol-d$_8$ buffer containing 1 M NaCl was added on top, without disturbing the pellet. The sample was incubated overnight at 4 ºC. Excess glycerol solution was removed, and the samples were transferred to 1.9 mm



rotors for subsequent DNP experiments. Final concentrations of 4.3, 22.8, and 28.2 mM were measured using a Bruker EMXnano benchtop EPR spectrometer directly on the packed DNP-NMR rotors prior to DNP experiments.

## 2.2 MAS NMR spectroscopy

DNP-enhanced MAS NMR spectra of CA tubular assemblies were acquired in the Bruker Billerica laboratories on an Avance III-HD SSNMR spectrometer equipped with a 1.9-mm triple-resonance low temperature MAS probe. At 14.1 T, the Larmor frequencies were 600.080 MHz ($^1$H), 150.905 MHz ($^{13}$C) and 60.813 MHz ($^{15}$N). The microwave (MW) frequency was 395.18 GHz and the MW irradiation generated by a second-harmonic gyrotron, which delivered 13.8 W of power at the sample. The measurements were performed at 120 K, and the sample temperature was calibrated using KBr (Thurber and

Tycko, 2009). All spectra were acquired at the MAS frequency of 24 kHz, controlled by a Bruker MAS3 controller. The typical 90º pulse lengths were 1.5 µs ($^1$H) and 3 µs ($^{13}$C). The $^1$H-$^{13}$C cross-polarization was performed with a tangential amplitude ramp on $^1$H with the center of ramp Hartmann-Hahn matched at the first spinning sideband; the carrier frequency on $^{13}$C was set to 100 ppm; the CP contact time was 2 ms. The $^{13}$C DANTE (Bodenhausen et al., 1976) pulse length was 0.05 or 0.1-µs. The DANTE interpulse delay was set to 1 or 2 rotor cycles. The 2D $^{13}$C-$^{13}$C CORD (Hou et al., 2013) spectra mixing time was

20 ms, corresponding to 480 rotor periods.

All spectra were processed in TopSpin 4.0 and analyzed in TopSpin 4.0 or NMRFAM-Sparky (Lee et al., 2015).

## 3 Results and discussion

$^{13}$C CPMAS and DPMAS NMR spectra of 5F-Trp,U-$^{13}$C,$^{15}$N CA tubular assemblies acquired with a recycle delay of 10 s are displayed in Figure 1a and 1b, respectively. The control spectra shown in the bottom traces, recorded with the

microwave power turned off, reveal that the addition of biradical gives rise to concentration-dependent signal intensity loss. While this effect is modest in the CPMAS experiment for the sample prepared with 4.3 mM AMUPol, in the DPMAS experiments the signal is attenuated considerably for the three biradical concentrations. Turning on microwaves at 13.8 W output power, a level sufficient to saturate the CE DNP mechanism (Lu et al., 2019), results in clear DNP signal enhancements. These enhancements in CPMAS experiments are large (76-fold), positive, and independent of the biradical concentration. This

is not the case for the DPMAS spectra, where the enhancements are much smaller: 4.6-fold (4.3 mM AMUPol), 3.2-fold (22.8 AMUPol), and 4.8-fold (28.2 mM AMUPol). Surprisingly, the DNP enhancement is negative for the sample prepared with 4.3 mM AMUPol, while those for the samples containing 22.8 and 28.2 mM AMUPol are positive. In addition, unexpectedly, while the spectral resolution of the non-enhanced and DNP-enhanced CPMAS spectra for these three samples and the DPMAS spectra of the 4.3 mM AMUPol sample are similar, the lines are dramatically broadened in the DNP-enhanced DPMAS spectra

of samples containing high AMUPol concentrations, 22.8 and 28.2 mM.

To gain further understanding of the origins of signal enhancements and spectral resolution in the three samples, we recorded DNP-enhanced CPMAS and DPMAS buildup profiles, varying the buildup times, Tb, from 10 µs to 64 s. Profiles



for different functional groups (carbonyl, aromatic, Cα, Cβ/Cγ, and Ile methyl groups) are displayed in Figures 2 and S1 of the Supporting Information. For CPMAS experiments, the signals are positive, and the buildup times become shorter as the

biradical concentration increases, as expected.

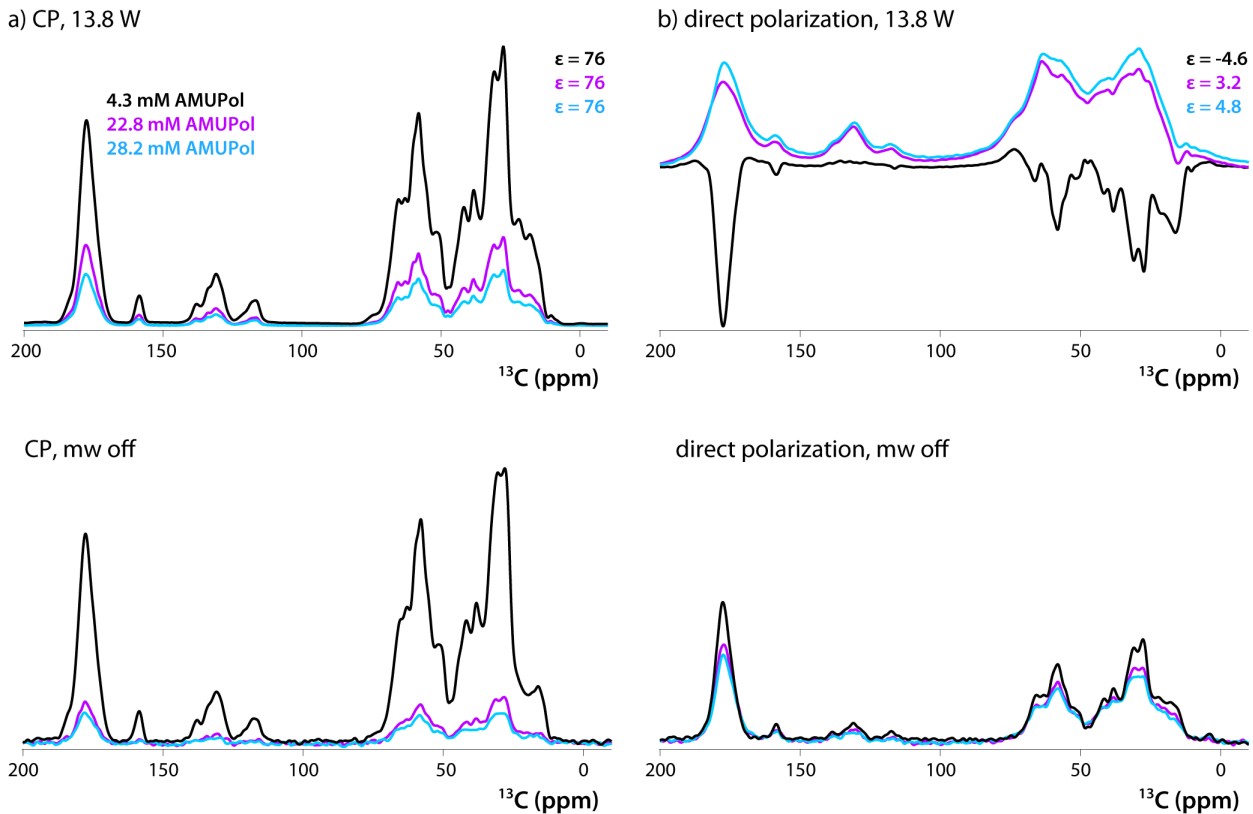

**Figure 1:** $^{13}$C DNP-enhanced CP (a, top), DNP-enhanced direct polarization (b, top), CP (a, bottom), and direct polarization (b, bottom) MAS NMR spectra of 5F-Trp,U-$^{13}$C,$^{15}$N CA tubular assemblies in the presence of 4.3 mM AMUPol (black traces), 22.8 mM AMUPol (magenta traces), and 28.2 mM AMUPol (blue traces). The non-DNP enhanced CPMAS spectra (bottom panels) are direct
intensities. For DNP-enhanced spectra (top panels) signal enhancements are indicated. The spectra were acquired at 14.1 T (150.96 MHz $^{13}$C Larmor frequency) at a MAS frequency of 24 kHz and 120 K. The recycle delay was 10 s.

The DPMAS DNP signal buildup profiles, on the other hand, are surprising. As shown in Figure 2a (middle panel), for the 4.3 mM AMUPol containing sample, the signals for the Ile methyl groups (shown as green and red curves) are always negative but their enhancement increases with buildup time, indicating that the polarization transfer proceeds via the
SCREAM-DNP pathway (Aladin and Corzilius, 2019). At the same time, the signals associated with other functional groups are small, positive, and build up quickly within the first 1-2 s. At longer times, negative intensity signals appear and build up, reaching the maximum negative enhancement of 6.4-fold at Tb = 64 s. Heteronuclear decoupling has no effect on these time dependencies: as shown in Figure 2a (bottom panel), turning the decoupling off only results in broadening of the signals but no sign inversion is seen. Furthermore, as shown in Figure 2c, at short buildup times, the positive signals are very broad, while







**Figure 2: Buildup profiles for $^{13}C$ signals in DNP-enhanced spectra of tubular assemblies of 5F-Trp,U-$^{13}C$,$^{15}N$ tubular CA assemblies containing a) 4.3 mM AMUPol, and b) 28.2 mM AMUPol. Top, CPMAS spectra; middle, direct polarization spectra acquired with decoupling; bottom, direct polarization spectra acquired without decoupling. Signals corresponding to different functional groups are color coded with the corresponding chemical shifts are displayed on the bottom right: carbonyl (177.4 ppm, black filled circles), aromatic (159.0, black open circles), Cα (66.8 ppm, magenta, 58.4 ppm, blue), Cβ/Cγ (28.3 ppm, gold), Ile methyl groups (15.5 ppm, red, 14.9 ppm, green). The buildup times and the maximum signal enhancements are indicated in each panel. c) DNP-enhanced DPMAS spectra of 5F-Trp,U-$^{13}C$,$^{15}N$ tubular CA assemblies containing 4.3 mM AMUPol recorded with different recycle delays: 500 ms (blue), 750 ms (grey), 1 s (magenta), 2 s (blue), 4 s (gold), 8 s (red), 16 s (green), 64 s (purple). The spectra were acquired at 14.1 T (150.96 MHz $^{13}C$ Larmor frequency) at a MAS frequency of 24 kHz and 120 K.**

the negative signals arising at long buildup times are narrow. Taken together, this suggests that at short buildup times for the sample with 4.3 mM AMUPol, the SCREAM-DNP pathway is operational for the Ile methyl groups while direct DNP transfer



occurs for the remainder of the functional groups. At buildup times exceeding 4 s, SCREAM-DNP becomes the dominant pathway throughout, yielding only negative signals and overall larger enhancements.

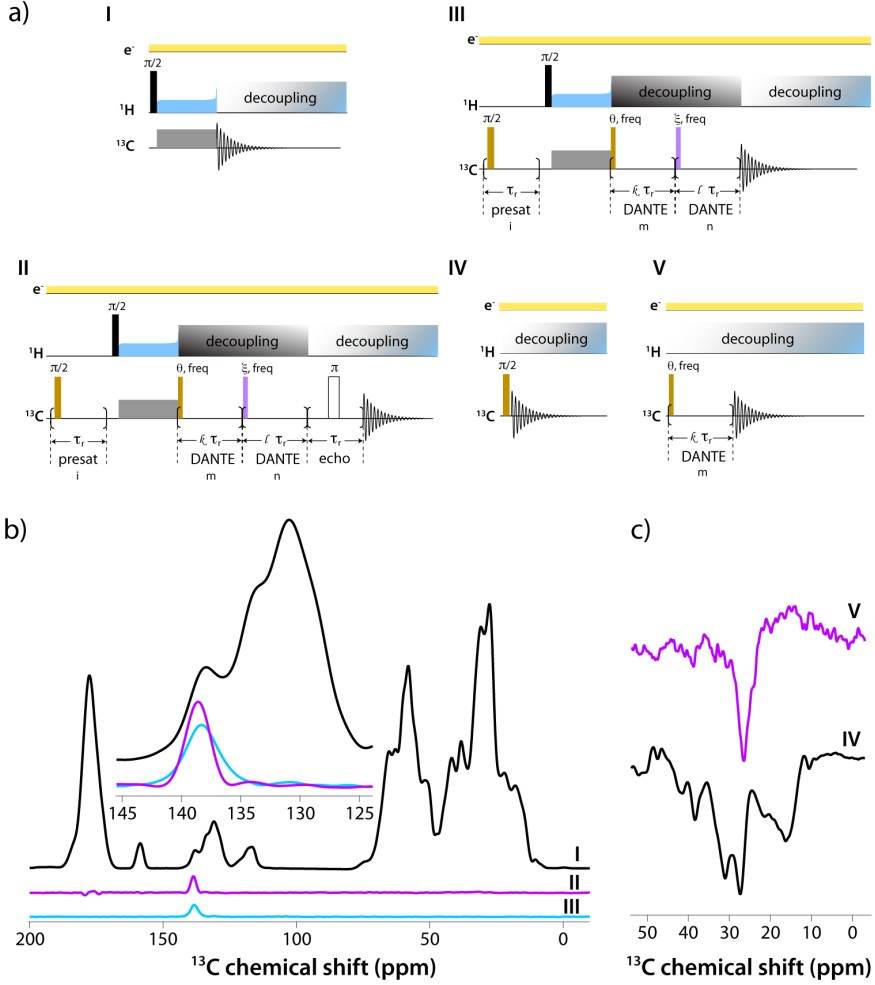

**Figure 3: a) Pulse sequences for DNP-enhanced MAS NMR experiments: CPMAS (I), CPMAS with presaturation, selective DANTE flip and readout, and Hahn echo detection (II), CPMAS with presaturation, selective DANTE flip and readout (III), direct polarization (IV), and direct polarization with selective DANTE excitation (V). b), c) DNP-enhanced spectra of 5F-Trp,U-$^{13}$C,$^{15}$N tubular CA assemblies, in the presence of 4.3 mM AMUPol. b) CPMAS-based spectra acquired pulse sequences I, II, and III are shown in black, magenta, and blue, respectively. The inset is an expansion around the aromatic region where the DANTE excitation pulse was applied (139 ppm). The homogeneous line width is 2.4 ppm (362 Hz). c) Direct polarization spectra acquired using pulse sequences IV and V are shown in black and magenta, respectively. The homogeneous line width is 3.7 ppm (559 Hz). The spectra were acquired at 14.1 T (150.96 MHz $^{13}$C Larmor frequency) at a MAS frequency of 24 kHz and 120 K.**

For the sample prepared with 28.2 mM AMUPol, SCREAM-DNP polarization transfer occurs for the Ile methyl groups, with negative signals building up until 2 s, after which the signal intensities decrease with the sign remaining negative. (This is different from the corresponding signal behavior in the 4.3 mM AMUPol sample, where the negative intensity builds up





until 64 s, see above.) Interestingly, all other functional groups are polarized via a direct pathway, with no evidence of SCREAM-DNP polarization occurring up to 16 s buildup time. The signals remain broad at all buildup times.

Intrigued by the relatively narrow lines in both CPMAS and DPMAS spectra acquired with 10 s recycle delay for the sample prepared with 4.3 mM AMUPol, we assessed the homogeneous line widths in each case. To this end, we performed
selective-excitation experiments using DANTE pulse trains, whose performance in terms of selectivity was optimized experimentally. The pulse sequences are shown in Figure 3a. Three CP-based experiments were performed: I) conventional CPMAS as a control; II) an experiment with $^{13}$C signal presaturation preceding CP (to eliminate any residual magnetization), followed by CP, double DANTE flipback/readout pulse train, and Hahn echo; and III) an experiment with $^{13}$C signal presaturation preceding CP (to eliminate any residual magnetization), followed by CP, double DANTE flipback/readout pulse
train. The corresponding spectra are displayed in Figure 3b. It is clear that experiment II has the highest selectivity, and the peak width is 2.4 ppm (362 Hz).

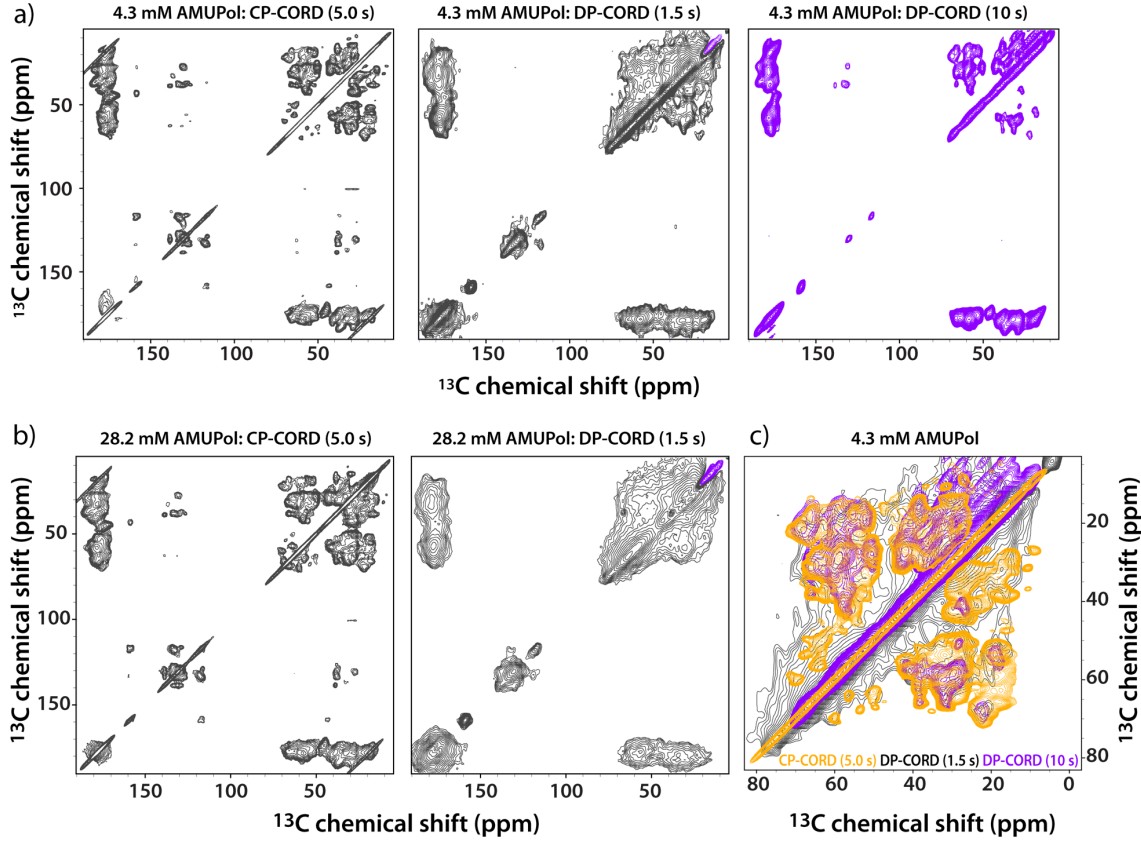

**Figure 4: DNP-enhanced CP-CORD (left) and DP-CORD (middle and right) spectra of 5F-Trp,U-$^{13}$C,$^{15}$N tubular CA assemblies in the presence of 4.3 mM AMUPol (a) and 28.2 mM AMUPol (b). The recycle delay in the CP-CORD spectra was 5 s, in the DP-CORD**
**spectra 1.5 s (middle) and 10 s (right). The CORD mixing time was 20 ms. The positive and negative signals are shown in black and magenta, respectively. c) The superposition of DNP-enhanced spectra shown in a): CP-CORD (gold), DP-CORD with a recycle delay of 1.5 s (black), and DP-CORD with a recycle delay of 10 s (magenta). The spectra were acquired at 14.1 T (150.96 MHz $^{13}$C Larmor frequency) at a MAS frequency of 24 kHz and 120 K.**





We also recorded two DP-based experiments: IV) a control with non-selective excitation, and V) a selective DANTE-excitation spectrum. Both are shown in Figure 3c. The peak width in the DANTE-excitation spectrum is 3.7 ppm (559 Hz), considerably broader than in the CP-based data sets.

Recognizing that the apparent broader line widths in the above 1D selective-excitation experiments may be associated with different sites in the uniformly $^{13}$C labeled protein, we performed 2D CP-CORD and DP-CORD experiments on the samples prepared with 4.3 mM and 28.2 mM AMUPol. The corresponding spectra are displayed in Figure 4a and b, respectively. The CP-CORD spectra acquired with a recycle delay of 5 s (left panels) are relatively well resolved with line widths of the individual resonances ranging from 0.8 to1 ppm and 1.4 to 1.8 ppm for 4.3 and 28.2 mM AMUPol containing samples, respectively.

In contrast to the relatively narrow lines in the CP-CORD spectra, the resolution in the DP-CORD spectra with a 1.5 s recycle delay is very poor for both samples (middle panels of Figure 4a and b), with SCREAM-DNP serving as the polarization transfer pathway for Ile methyl groups (negative peaks, purple), and direct transfer being operational for the rest of the functional groups (positive peaks). It is apparent that paramagnetic line broadening is severe, and the spectra appear to report selectively on surface residues. For the few resolved cross peaks, the line widths are on the order of 3 ppm in both spectra.

The DP-CORD spectrum with a 10 s recycle delay for the sample containing 4.3 mM AMUPol is shown in Figure 4a, right panel. Interestingly, the spectral resolution is reasonably good, with the with line widths of the individual resonances ranging from 1.5-1.8 ppm. While many cross peaks superimpose well on those in the CP-CORD spectra, a number of resonances that are present in DP-CORD data sets are missing in CP-CORD spectra, and vice versa (Figure 4c).

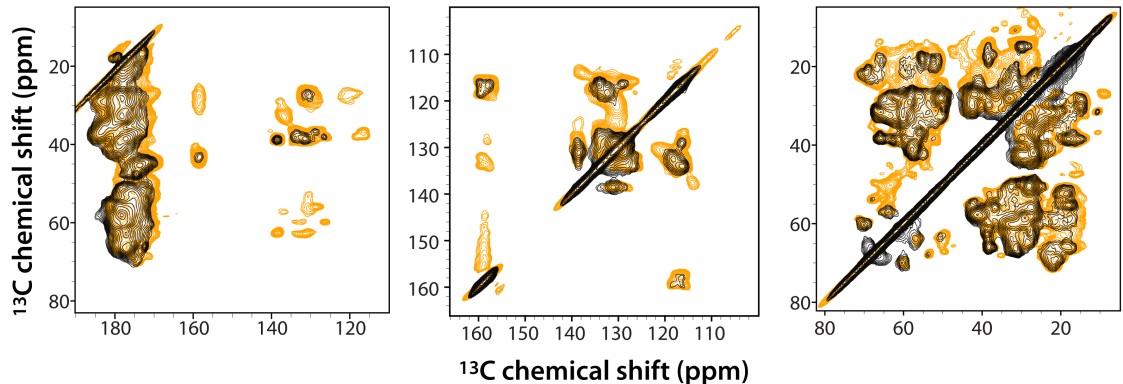

**Figure 5: Superposition of DNP-enhanced CP-CORD spectra of 5F-Trp,U-$^{13}$C,$^{15}$N tubular CA assemblies in the presence of 4.3 mM AMUPol (gold) and 28.2 mM AMUPol (black). Expanded regions of the spectra around carbonyl-aliphatic/aromatic-aliphatic (left), aromatic (middle), and aliphatic (middle) are shown. The recycle delay was 5 s and the CORD mixing time was 20 ms. The spectra were acquired at 14.1 T (150.96 MHz $^{13}$C Larmor frequency) at a MAS frequency of 24 kHz and 120 K.**

The superposition of DNP-enhanced CP-CORD spectra of samples prepared with 4.3 mM AMUPol (gold contours) and 28.2 mM AMUPol (black contours) is displayed in Figure 5. As can be appreciated, the overall spectral resolution is comparable (if somewhat lower in the 28.2 mM AMUPol containing spectrum), and many cross peaks are missing at the higher AMUPol concentration.



## 4 Conclusions

Using tubular assemblies of HIV-1 CA protein as a model system, we discovered that all three DNP polarization transfer pathways – indirect, direct, and SCREAM-DNP – are simultaneously active and can be emphasized or selected by carefully choosing specific sample conditions and/or experimental set-ups. While the indirect DNP pathway results in the highest signal enhancements and narrowest lines, direct DNP-based experiments permit the identification of surface sites in close proximity to the radical in these tubular assemblies. Taken together, our results also suggest that for attaining high enhancements and spectral resolution simultaneously, there may be no advantage to using high biradical concentrations: for our current sample, 4.3 mM AMUPol yielded the highest DNP signal enhancements and the best resolution. Conversely, high biradical concentrations can be employed to selectively bleach surface signals, allowing one to focus on other sites. We envision that our findings may provide valuable guidance for structural investigations of other biological assemblies by DNP MAS NMR.

## 5 Acknowledgments

This work was supported by the National Institutes of Health (NIH Grant P50AI150481) and is a contribution from the Pittsburgh Center for HIV Protein Interactions. We acknowledge the support of the National Institutes of Health (NIH Grant P30GM110758) for the support of core instrumentation infrastructure at the University of Delaware. We thank Manman Lu and Mingzhang Wang for DNP sample preparations.

## 6 Author contribution

T.P., I.V.S., and A.M.G. conceived the study. I.V.S., T.P., and C.M.Q. performed the NMR experiments. T. P. and I. V. S. analyzed the data. T.P. took the lead in writing the manuscript. All authors discussed the results and contributed to the manuscript preparation.

## 6 Competing interests

The authors declare no competing interests.

## 8 Supporting information

Buildup profile for $^{13}$C signals in DNP-enhanced CPMAS spectra of tubular assemblies of 5F-Trp,U-$^{13}$C,$^{15}$N CA containing 22.8 mM AMUPol.



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
