# Peer review of "Competing Transfer Pathways in Direct and Indirect Dynamic Nuclear Polarization"

_Magnetic Resonance, 2021_

## Referee Comment (RC2)

Polenova, Gronenborn and co-workers provide a detailed investigation into the competing transfer pathways active in biomolecular under MAS DNP. Here, three different pathways are currently known, namely CP-based indirect DNP, direct DNP, as well as a spontaneous contact between 1H and 13C magnetization mediated through heteronuclear cross-relaxation (SCREAM-DNP). The authors excellently provide a short review about these pathways and introduce the reader to the topic under investigation.

Understanding the interplay between the three transfer pathways is of crucial importance to successfully apply MAS DNP towards different questions by choosing the optimal conditions for the task at hand. While some control may be exerted by choice of the pulse sequence, the disentanglement of the pathways is not trivial. In this regard, this work gives very useful information about the dynamics of each transfer pathway and their dependence on the polarizing agent concentration. Furthermore, line broadening is discussed for various distinct resonances under the different experimental conditions.

The manuscript is well written and of good scientific quality. All experimental data is well presented and documented, and reproduction seems possible given the provided information. The content is of general interest to a broad magnetic resonance community and therefore suitable for publication in MR. Before publication, however, I ask the authors to comment on several specific points and perform minor revisions of the manuscript accordingly.

In line 40, the authors state that "modest gains have been detected for membrane proteins with $\varepsilon$=4-10 (Wylie et al., 2015)". This statement reads like a general observation, however, the referenced work covers a rather special case of membrane proteins labeled with single nitroxide tags which come into dipolar contact. In my experience, DNP enhancement of 40-60 can be routinely achieved for membrane proteins.

The following sentence ("Recently, large DNP signal…") is confusing. The first part deals with impregnated microcrystalline histidine, the second part is in no way generally applicable and seems to be specifically limited to the highest available field. Even in this regard, the sentence is misleading because it implies that no soluble polarizing agents are available for biological systems.

The authors use "build-up time, Tb" interchangeably for both the time constant as well as the experimentally chosen polarization time period (c/f lines 127, 129, 142, and Figure 2). These two different parameters have to be clearly distinguished by an unambiguous choice of symbol and naming.

In Figure 2, it is not clear, which graphs the subpanels abc are referring to. Also, in the lower left graph, epsilon(-) seems to be missing a negative sign (-4).

When discussing the build-up dynamics in Figure 2, it should be clarified that SCREAM-DNP magnetization is emerging from the methyl groups and the spreading through the 13C network by spin diffusion. This explains the quick inversion of Ile resonances, and the delayed response of the other resonances.

On the bottom of page 5, it is stated that "Heteronuclear decoupling has no effect…". I am wondering by what means turning off the decoupling is expected to effect a sign inversion? Heteronuclear decoupling is used when the 13C magnetization is already in the transverse plane, so it is unclear to me how this can influence the sign of polarization. For SCREAM-DNP 1H saturation during the build-up period mostly destroys the incoherent pathway, but this is independent from decoupling. This part should be revised, it should either be explained why decoupling may be expected to change the outcome of the experiment, or it should be clarified if indeed decoupling is mistaken for saturation.

In line 204, "with the with line…" seems to contain an additional "with".

---

## Author Comment (AC2)

*The manuscript is well written and of good scientific quality. All experimental data is well presented and documented, and reproduction seems possible given the provided information. The content is of general interest to a broad magnetic resonance community and therefore suitable for publication in MR. Before publication, however, I ask the authors to comment on several specific points and perform minor revisions of the manuscript accordingly.*

We thank the reviewer for the positive assessment of our work and for helpful suggestions on how to improve our manuscript. The point by point response to the reviewer's comments is below.

*In line 40, the authors state that "modest gains have been detected for membrane proteins with ε=4-10 (Wylie et al., 2015)". This statement reads like a general observation, however, the referenced work covers a rather special case of membrane proteins labeled with single nitroxide tags which come into dipolar contact. In my experience, DNP enhancement of 40-60 can be routinely achieved for membrane proteins.*

We have thoroughly revised the paragraph in question to more clearly delineate parallel discussions of DNP at high magnetic field and DNP in biological systems. We have added a wider range of enhancement factors, 20-100, that can be achieved at lower field based on an overview of available literature – these numbers can vary greatly depending on the precise details of the sample.

*The following sentence ("Recently, large DNP signal…") is confusing. The first part deals with impregnated microcrystalline histidine, the second part is in no way generally applicable and seems to be specifically limited to the highest available field. Even in this regard, the sentence is misleading because it implies that no soluble polarizing agents are available for biological systems.*

We agree that the original sentence was confusing and have revised it.

*The authors use "build-up time, Tb" interchangeably for both the time constant as well as the experimentally chosen polarization time period (c/f lines 127, 129, 142, and Figure 2). These two different parameters have to be clearly distinguished by an unambiguous choice of symbol and naming.*

We agree with the reviewer and have changed the notations for the time constant of polarization buildup and the experimentally chosen polarization transfer time period to $T_B$ and the "recycle delay", respectively.

*In Figure 2, it is not clear, which graphs the subpanels abc are referring to. Also, in the lower left graph, epsilon(-) seems to be missing a negative sign (-4).*

We have revised the figure caption to clarify what the individual subpanels are referring to and added a negative sign to epsilon(i) to the bottom left panel.

*When discussing the build-up dynamics in Figure 2, it should be clarified that SCREAM-DNP magnetization is emerging from the methyl groups and the spreading through the 13C network by spin diffusion. This explains the quick inversion of Ile resonances, and the delayed response of the other resonances.*

Thank you for this suggestion; we have added this clarification to the discussion.

*On the bottom of page 5, it is stated that "Heteronuclear decoupling has no effect…". I am wondering by what means turning off the decoupling is expected to effect a sign inversion? Heteronuclear decoupling is used when the 13C magnetization is already in the transverse plane, so it is unclear to me how this can influence the sign of polarization. For SCREAM-DNP 1H saturation during the build-up period mostly destroys the incoherent pathway, but this is independent from decoupling. This part should be revised, it should either be explained why decoupling may be expected to change the outcome of the experiment, or it should be clarified if indeed decoupling is mistaken for saturation*

The intention here was to test whether, by simply decoupling H from C NMR spins, the heteronuclear cross-relaxation effect (SCREAM-DNP) can be suppressed. Clearly, it cannot, nor would we necessarily have expected it to be. We have therefore removed the reference to sign inversion but left the note about decoupling, since we feel it is nonetheless important to state that it has no effect on the SCREAM-DNP mechanism, consistent with predictions. The revised sentence is: "As expected, heteronuclear decoupling has no effect on these time dependencies, as shown in Figure 3a (bottom panel): turning the decoupling off only results in broadening of the signals."

*In line 204, "with the with line…" seems to contain an additional "with"*

We corrected this typo.

---

## Author Comment (AC3)

**Reviewer 3:**

*On the whole, the study is very interesting and the following remarks are meant to be part of a final ironing process. ... The paper is certainly worthwhile publishing, and I enjoy reading it; however, it can do with a lot of clarifications to improve readability.*

We thank the reviewer for the positive assessment of our work and for helpful suggestions on how to improve our manuscript. The point by point response to the reviewer's comments is below.

*The authors claim that 13.8 watts of MW were applied at the sample. Where was that measured, and how was that measured? This should be included in the experimental. Was it measured in front of the sample or in the sample? What exactly means 'at' in this case?*

The MW power was measured at the waveguide entrance. We have revised the corresponding sentence of the Experimental section to: "The microwave (MW) frequency was 395.18 GHz and the MW irradiation generated by a second-harmonic gyrotron, which delivered 13.8 W of power, as measured at the waveguide entrance."

*The spectra shown in Fig. 1 were all recorded with a recycle delay of 10 s. The authors should briefly state why they choose to do the experiment in this way and not using appropriately chosen multiples of T1 which is very different for the samples, and which would lead to optimal signal-to-noise-ratios for the fast relaxing ones. More importantly, they should discuss the implications of choosing the same delay for all samples appropriately. Enhancements are given in the top row of Fig. 1a, and I wonder whether there is a typo somewhere. Since so different radical concentrations are used I would expect different depolarization effects, by the way. One sentence discussing their possible contributions and in general the change with radical concentration would be good.*
*The authors should also indicate the scaling factors between the displays of spectra in the top and bottom rows in Fig. 1. They cannot be the same, otherwise the statement that in all three cases E=76 is likely not correct.*

Thanks for alerting us to a typo and a mistake in Figure 1. On the bottom left panel, the traces for CP spectra of samples with 22.8 and 28.2 mM AMUPol were displayed with incorrect normalization. We have corrected this mistake as well as the typo in the signal enhancement values in the revised manuscript.

As to the choice of recycle delay, we have added the following explanation/justification to Results and Discussion:

"In conventional NMR, where a single variable ($T_1$) governs longitudinal spin relaxation, the recycle delay is generally simply chosen to maximize signal-to-noise ratio per unit time (e.g., $1.3*T_1$). The situation in DNP-NMR is significantly more complex: here, the recycle delay represents not only the longitudinal relaxation period but also the polarization buildup time period, since microwaves are always on throughout the experiment. If multiple DNP mechanisms are involved, each may have a different polarization buildup time constant ($T_B$), further varying by site on the molecule. The relationship between the experimental recycle delay and $T_B$ governs the relative contributions of the various mechanisms. As a result, the DNP buildup profiles provide unique insight into the complex interplay of DNP effects."

*It should always clearly be stated which pulse sequences were used; names are not enough. Fig. 3 is good to have, but where is the SCREAM sequence, where are the CORD sequences (yes, they are easy), which sequence is used for which spectra in Fig. 1, 2 and 4, etc. For the outside reader it is very difficult to understand what TB, TB+ and TB- refers to, there is nothing like this in the pulse sequences of Fig. 3 and there is also no description in the experimental. Furthermore, the letters in the pulse sequences of Fig. 3 are far too small, especially those for the DANTE delay. It requires quite some magnification to see that it is Tr. Elderly persons who need to print the manuscript will not see anything.*

SCREAM is not an NMR sequence but rather a DNP mechanism, which is explained in the text. The SCREAM-DNP effect is clearly seen in the 1-pulse (direct polarization experiments). In regard to the SCREAM sequence, the reviewer appears to be referring to a difference experiment introduced by Corzilius and co-workers, where a spectrum acquired by DP (a combination of direct and indirect transfer pathways) was subtracted from the DP spectrum with saturation pulses applied to suppress the indirect transfers, hence yielding pure spectrum from the indirect pathway due to the methyl group dynamics. The purpose of the current study was to discern the competing transfer pathways, and therefore there was no need for such a difference experiment: the SCREAM

pathway is clearly seen in a DP experiment, Figure 4a (trace IV).

We have compiled all pulse sequences used in this work into a separate figure (Figure 1 of the revised manuscript) and indicated clearly in the figure captions to Fig. 2-6 which sequence was used for which experiment. We increased the font size in Figure 2 of the revised manuscript, so that the individual letters can be seen clearly.

*Fig. 4 reports CP-CORD spectra whereas this name does not appear in Fig. 3 nor in the experimental part of the paper. There is no reference where I would expect it. Again, its needs to be stated to which pulse sequence Fig. 4 correlates to, and what the numbers mean above the spectra. By the way, they are very misleading, they could be mistaken for the CORD mixing time by an unexperienced reader. Better write RD= and define somewhere in the paper or in the legend.*
There are no CP-CORD spectra anywhere in Figure 3; none of the spectra presented in Figure 3 of the original submission used CORD or any other C-C mixing periods. We added a reference to the CORD paper in the Experimental section. We have placed all pulse sequences in Figure 1 of the revised manuscript, including CP-CORD and DP-CORD sequences, as discussed above.

*It is very instructive to see spectra recorded with 4.3 mM Amupol only, it is an important point of the paper, but it is always tricky to compare contour plots. To me it is obvious that spectra are different yet comparing one or two selected cross sections would be probably wise.*
Thank you for this suggestion. We have added several selected 1D traces from the 2D spectra to the figure so that the differences are clear.

*In the conclusion section it is said that 4.3 mM Amupol yielded the highest DNP signal enhancements. In stark contrast, Fig. 1 announces exactly the same enhancement for all three concentrations. Furthermore, the statement needs probably some seasoning, since there was no T1-optimized relaxation delay employed, and the S/N would be better with appropriate choices for the samples with higher Amupol concentrations. On the other hand, the better resolution in the spectra for the sample 4.3 mM Amupol does its job, too. Resolution is good, yes. To me it looks like similar T1 effects lead to the appearance of narrow signals in those direct polarization experiments recorded with longer relaxation delays. Maybe this should also be discussed appropriately.*
The same enhancement factors was a typo, which we have now corrected. As mentioned in previous comments, finding a $T_1$-optimized relaxation delay for DNP spectra is not quite as straightforward as it might be in conventional NMR: each DNP mechanism must be considered and weighted. For HIV-1 CA tubular assemblies discussed in this work, we find that the optimal balance between sensitivity and resolution is achieved for 4.3 mM AMUPol concentration and recycle delay of 10 s.